# Compost Fertilization in Organic Agriculture—A Comparison of the Impact on Corn Plants Using Field Spectroscopy

**Martin Strenner \*, Lucie Chmelíková and Kurt-Jürgen Hülsbergen**

Chair of Organic Agriculture and Agronomy, TUM School of Life Sciences, Technical University of Munich, 85354 Freising, Germany
\* Correspondence: martin.strenner@tum.de

**Featured Application: Potential use of a fast and non-destructive method to detect and optimize organic fertilization.**

**Abstract:** To protect the environment and reduce the impact of fertilizing on climate change, a tailored fertilization according to the needs of the plants becomes more and more important. In organic farming, the main sources of nitrogen (N) are organic fertilizers, such as compost and farmyard manure. In conventional agricultural systems, various plant sensors have already proven that they can guide efficient fertilization. Since these sensors can record the growth of the biomass growth, they can also be used in organic farming to detect the influence of various organic fertilizers on the growth of corn plants. In a field trial established in 2017, eight different organic fertilizers (biogas fermentation residue, matured farmyard manure, fresh and matured organic waste compost, fresh and matured green compost, and microbial carbonized compost) were used and applied in two different amounts (120 and 240 kg N ha$^{-1}$) along with an unfertilized control variant. The fertilization was applied according to the current version of the German fertilizer ordinance. In 2021, sensor measurements of the corn plants were taken at six different stages of development (from BBCH 13 to BBCH 35) with a hand-held field spectrometer using the REIP vegetation index. The measurements showed that the unfertilized control variant could be reliably recorded. Furthermore, even at early growth stages the measurements showed the differences between different composts. The results presented in this study show that sensor measurements using vegetation indices reacted sensitively to organic fertilization and can be a tool for farmers to support their decision for an adequate fertilization strategy.

**Keywords:** corn; fertilization; compost; organic fertilizer; optical sensor measurements; vegetation index

## 1. Introduction

As in every crop production system, an optimal supply of nutrients for plant production is essential. In particular, nitrogen (N) is an elementary nutrient component for optimal plant growth. Plant N concentrations below an optimal value can lead to significant yield losses [1–3]. High doses of applied nitrogen will lead to a loss of N in different forms, i.e., ammonia, nitrous oxide emissions from the soil, and nitrate leaching into the groundwater [4–7]. This loss leads to negative effects to the environment, as well as to a contribution to climate change [8–10]. However, this does not only apply to conventional farming systems. Even in organic farming, there are risks of N losses, such as nitrous oxide emissions after ploughing clover grass or green manure [11,12].

In comparison to conventional farming, the input of nutrients in organic farming is relatively low, especially in terms of nitrogen [13–16]. In organic farming, a large source of nitrogen input, besides compost, is N$_2$ fixation because of the symbiotic interaction of legumes and rhizobia. This provides substantial amounts of N supply to the plants and the soil as well. Furthermore, it reduces the need for additional N fertilizers [17]. However,

fertilization only based on $N_2$ fixation does not suite the needs of the plants. A possible solution for this lack of nutrients is to close the cycle of nutrients. With regard to climate change and reducing the cost of fertilizer, the recycling of nutrients, especially nitrogen and phosphorus, is becoming more and more important [18,19]. One way to recycle nutrients is through the use of compost [20,21]. The use of compost also offers an optimization of nutrient supply for organic plant production.

Unlike mineral fertilization, the release of nutrients from organic fertilization is much slower [22]. In particular, N from compost is initially stored in the soil N pool and will be released in a plant-available form often after decades [23]. It is, therefore, much more difficult for the farmer to determine the effect of organic fertilizers on the crop and the soil as well. N mineralization, which produces a plant-available form of N, is essential for plants and depends on the soil as well as the weather conditions [24], but does not coincide with plant requirements [13]. In addition to the above-mentioned N mineralization, the release of soil-borne N shows no homogeneity within a heterogeneous field [25]. As the necessary amount of fertilized N is the difference between the required N of the crops minus the supply of soil-borne N, the challenge for organic farming is to know and decide which and how much organic fertilizer to use (in this study different composts were used as organic fertilizers) to best suit the needs of the crop. A conventional approach for researchers to gather reliable information about the optimal N fertilization is the use of multi-location and multi-annual field trials [3,26], to gather many soil and plant samples to analyze them based on their nutrient status. For an estimation of the N requirements only based on soil analyses, a high number, as well as a high areal resolution, of soil tests becomes necessary. This approach is time-consuming and not economic [27,28].

The growth and development of plants are dependent on the actual nutrient status. Therefore, plants themselves are the best indicators for estimating their nutritional status, i.e., N requirement and above-ground biomass [29–32]. The current N status can be determined easily using spectral sensors, which detect the light of different wavelengths reflected by the plants [33–36]. In addition, this method is non-destructive, which means that in field use it is not required to cut plant samples to analyze them afterwards in a laboratory to obtain information about their actual nutrient status. In conventional farming, non-destructive spectral sensing has already proved its ability to guide nitrogen fertilization in winter wheat [29,37]. Sensors that detect the reflectance usually show their results as vegetation indices. Common sensors on the market, such as the GreenSeeker (Trimble Inc., Sunnyvale, CA, USA) mostly show their measuring results using the normalized differentiated vegetation index (NDVI). This vegetation index is one of the oldest [38] and has serious problems in detecting the biomass and N uptake of plants at higher growth stages because of a significant saturation effect for various crop species [39–42]. As a result of this saturation effect of the NDVI, another vegetation index, the red-edge inflection point (REIP) [43], was recommended to detect the N status of plants [44,45]. In this study, we use the vegetation index REIP, which has shown a slight saturation effect only in the later growth stages of a crop [46,47] and has consistently shown a good correlation of the REIP vegetation index and above-ground N uptake of corn plants [45].

As described, optical sensors have already proven their ability to detect the biomass and nutrient status in conventional grown cereal plants. As corn (*Zea mays* L.) was in 2021 the most cultivated grain worldwide [48], and although conventional corn cultivation predominates in Germany, a clear expansion of organically grown corn can be observed [49]. Most of the studies using optical sensors to detect the nutrient status of plants in agriculture are made for conventional agricultural systems to guide fertilization. The aim of this study was to research the ability of optical sensors using the REIP vegetation index to detect differences in development of corn plants, where different composts fertilizers were used in an organic field trial.

## 2. Materials and Methods

The field trial was established in the northeast of Munich at the organic trial site of the agricultural research center of the Technical University of Munich near Freising-Weihenstephan (48.39742 N, 11.64324 E) as a long-term experiment in autumn 2017. The experiment on different organic fertilizers was established as part of the ProBio Project (Project title: Optimum production and agronomic use of compost from organic and green waste in organic farming, www.projekt-probio.de, accessed on 20 December 2022). The soil is a silty and sandy loam with good supply of nutrients for grain production. The average values from 2017 until 2021 at this location were 914.4 mm of precipitation with an average temperature of 9.4 °C. Figure 1 shows an overview of the average monthly precipitation and temperatures at the research location.

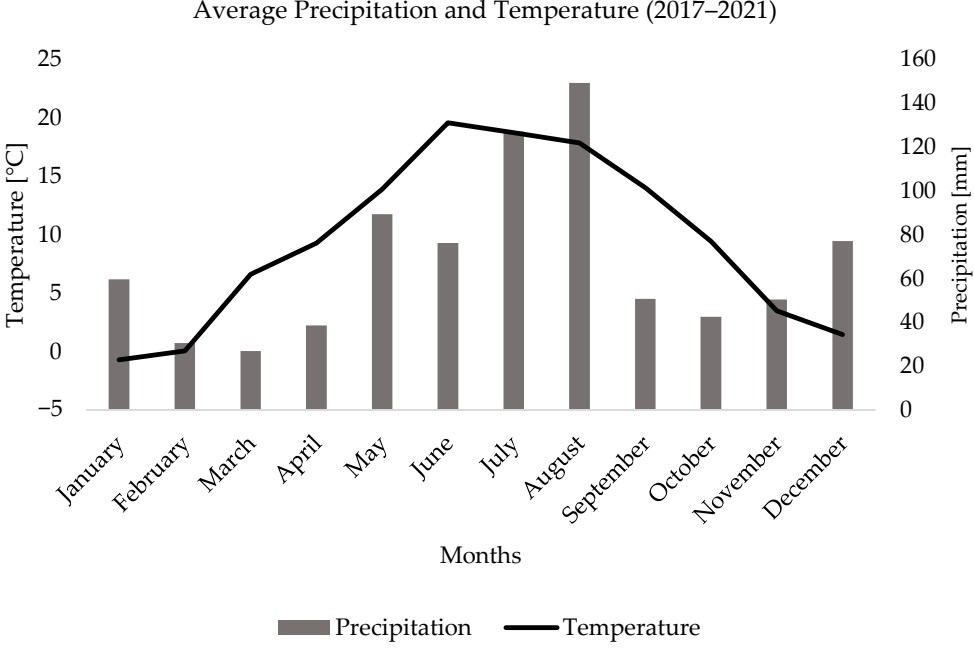

**Figure 1.** Average precipitation and temperature at research location (2017–2021).

In 2021 after sowing corn by the end of April for the growth of the corn plants in the months from May to August, the precipitation was higher than the mean value shown in Figure 1. In May the precipitation was, with 156 mm, nearly 60 mm above the mean value. The same could be observed for June with 140 mm of precipitation and, thus, being approximately 60 mm above the mean value. For July 2021, the precipitation was approximately 135 mm and within the mean values. In August 2021, there was 171 mm precipitation, which was 15 mm higher than the mean value. In 2021 for the recorded temperatures, May had an average temperature of 10 °C, which was approximately 2 °C colder than the mean values from 2017 to 2021. Whilst May 2021 was too cold, June 2021 was, with an average of 20 °C, approximately 1 °C too hot. July and August were, with recorded temperatures of 18 °C and 17 °C, respectively, within the mean values.

Before the experiment was established, soil samples were taken on all plots in order to record the initial humus and nutrient levels. For the compost trial, a fully randomized latin square design was used with four replicates. One plot was used for sensor measurements and harvesting, whilst the other was for biomass sampling at different growth stages. Each plot was 10 m long and 3 m wide, and in 2021 each plot consisted of four rows of corn. The row width was 75 cm and the plant density was 10 plants per m$^2$.

In the observation period, in early April 2021, before planting the corn and depending on their N content, seven different organic compost fertilizers were applied, along with an unfertilized control. Five of them had two different N rates of 120 kg N ha$^{-1}$ and 240 kg N ha$^{-1}$. The application rate and nutrient content of the different composts are

given in Table 1. The fertilization was conducted according to the German fertilizer ordinance in one application for three years.

**Table 1.** Applied N rates and nutrient content of the organic fertilizers (2021).

| Compost | Application Rate (kg N ha$^{-1}$) | Dry Matter (%) | N (%) | C/N (Ratio) | P$_2$O$_5$ (%) | K$_2$O (%) |
|---|---|---|---|---|---|---|
| Biogas fermentation residue | 120 | 23.6 | 1.65 | 26.5 | 1.12 | 2.78 |
| Farmyard manure (matured) | 120 | 22.2 | 1.86 | 17.9 | 0.81 | 4.10 |
| Organic waste compost (fresh) | 120 / 240 | 55.1 | 1.57 | 16.7 | 0.65 | 1.09 |
| Organic waste compost (matured) | 120 / 240 | 63.4 | 1.72 | 12.5 | 0.91 | 1.64 |
| Green compost (fresh) | 120 / 240 | 59.2 | 2.12 | 18.5 | 0.74 | 1.33 |
| Green compost (matured) | 120 / 240 | 56.9 | 1.44 | 15.7 | 0.59 | 1.07 |
| Microbial carbonized compost | 120 / 240 | 45.1 | 1.52 | 12.3 | 0.69 | 2.09 |

The composts we used were from organic farmers (biogas fermentation residues, matured farmyard manure and the microbial carbonized compost). The organic waste composts, as well as the green composts, came from compost producers. These producers are certified according to RAL (Deutsches Institut für Gütesicherung und Kennzeichung e.V., Bonn, Germany). The complete process of composting consists of three phases. The first is the mesophilic phase with moderate temperature. It lasts only a couple of days. The second is the thermophilic phase with high temperatures up to 70 °C. This phase lasts between a few days to several months. Finally, the third phase is the maturation of the compost, which can last several months [50]. Unlike matured compost, fresh compost is gathered after the end of the mesophilic phase and just before the thermophilic phase has begun. The microbial carbonized compost is a special case. Unlike conventional produced composts, microbial carbonized composts produced according to Walter Witte (Witte Bio Consult, Gernrode, Germany) are not mixed up and during composting the temperature within the pile of the compost should not exceed 60 °C. This manufacturing method should keep the carbon of the organic matter within the compost and should, therefore, avoid a loss of carbon into the atmosphere in the form of $CO_2$.

The crop management on the trial site was performed according to the organic farming guidelines of the German organic cultivation associations Bioland (Bioland e.V., Mainz, Germany) and Naturland (Naturland–Verband für Ökologischen Landbau e.V., Gräfelfing, Germany) by hoeing and harrowing.

Three times during the growing period, at growth stages BBCH 32, BBCH 49 and BBCH 85 [51], above-ground biomass samples were gathered. For each biomass sampling, 16 plants of two planting rows of each plot were harvested, which covered a sample area of 1.5 m². The samples were afterwards weighed and chopped. Then, a sub-sample was oven dried and weighed again to calculate above-ground dry-matter yield, followed by a conversion of the values into mass per ha. A part of the sample was grounded and afterwards the N content was analyzed using a varioMAX auto-analyzer (Elementar Analysensysteme GmbH, Hanau, Germany). Dry-matter yield and N concentration in dry matter were multiplied to compute N uptake into above-ground biomass.

Measurements of the reflected sunlight were made six times during the vegetation period, at growth stages BBCH 13, BBCH 14, BBCH 15, BBCH 32, BBCH 33, and BBCH 35.

The sensor device for detecting reflectance spectra was a HandySpec two-channel field spectrometer of tec5 AG (tec5 AG, Steinbach, Germany). It simultaneously recorded irradiation and reflection with a spectral resolution of 1 nm in a range from 400 nm to 850 nm. An optical fiber receiver within the device recorded the sunlight reflected by the vegetation and ground with a viewing angle of 25° to determine reflection. A diffuser plate on top of the device detected irradiance within a viewing angle of 180° as reference. To calculate spectral reflectance, the emission was divided by irradiance and corrected with an internal standard spectrum. The sensor device was held approximately 50 cm above the canopy in a horizontal position. Each measurement covered an area of 0.25 m². While walking through the plots, in each plot, seven readings were recorded and an average value was calculated. The measurements were made between approximately 11 a.m. and 1 p.m. to be sure the sun was approximately at its zenith.

Figure 2 shows the reflection curves of the corn plants recorded using the field spectrometer at the research location at growth stage BBCH 32. Table 2 shows how the REIP vegetation index was calculated and which wavelengths are needed.

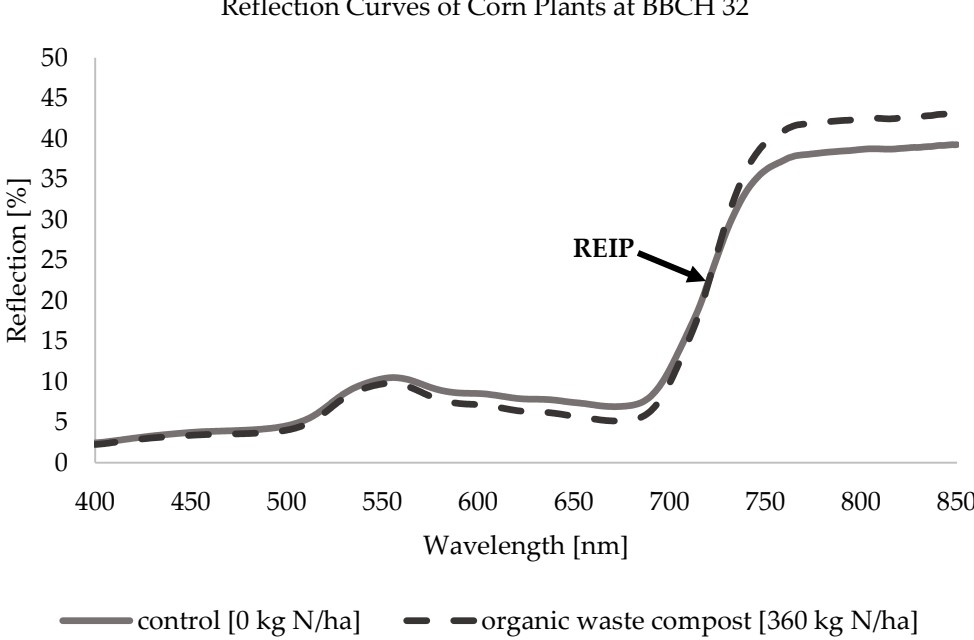

**Figure 2.** Typical reflection curves of corn plants at growth stage BBCH 32.

**Table 2.** Formula and reference of the REIP vegetation index.

| Vegetation Index | Formula | Reference |
| :---: | :---: | :---: |
| REIP | $700 + 40 \frac{0.5(R_{670}+R_{780}) - 700}{R_{740} - R_{700}}$ | [43] |

Analysis of variance (ANOVA) was calculated with the statistical software package SPSS 23.0 (IBM Inc., Armonk, NY, USA) using the general linear model and Tukey-b tests for post-hoc analysis.

## 3. Results

In the observation period in 2021, at the first measurement date at growth stage BBCH 13 (Figure 3) the REIP-values were highest in the variants with fresh organic waste and green compost with high application rates, as well as with biogas fermentation residues. The corn plant was at that time in the three-leaf stage and had developed therefore only a little biomass. The N uptake from the soil was correspondingly low.

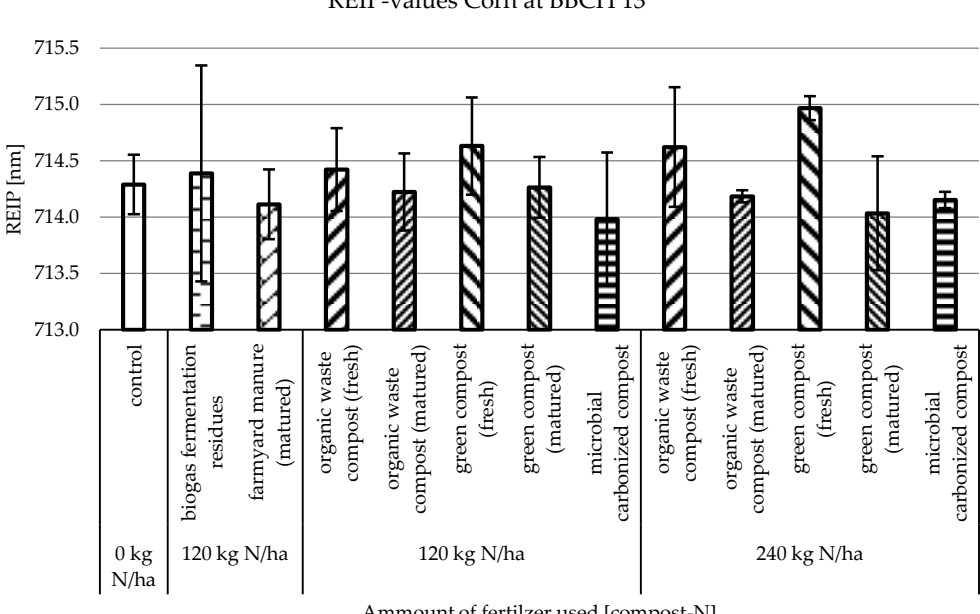

**Figure 3. Measured** REIP-values at BBCH 13.

At the four-leaf stage at BBCH 14 (Figure 4), these differences became statistically significant (ANOVA and post-hoc test with Tukey-b).

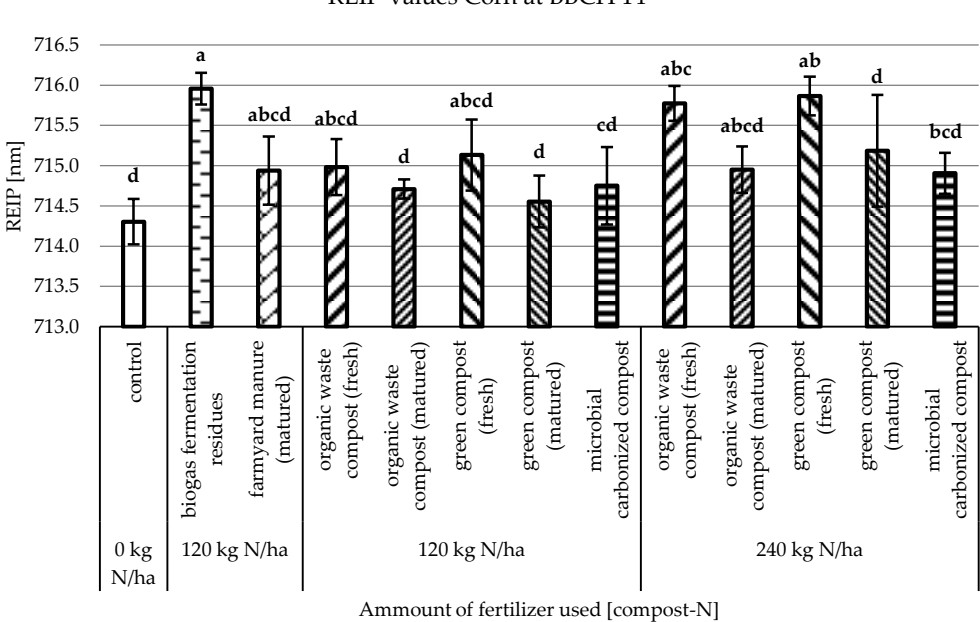

**Figure 4.** Measured REIP-values at BBCH 14 (variants with the same letters do not differ significantly at 95% level; ANOVA with Tukey-b test).

At growth stage BBCH 32 (Figure 5), the REIP-values looked different. The highest values were measured in the variants with matured organic waste compost and microbial carbonized compost with the high application rate. In contrast, the values for the fresh compost and biogas fermentation residue had decreased.

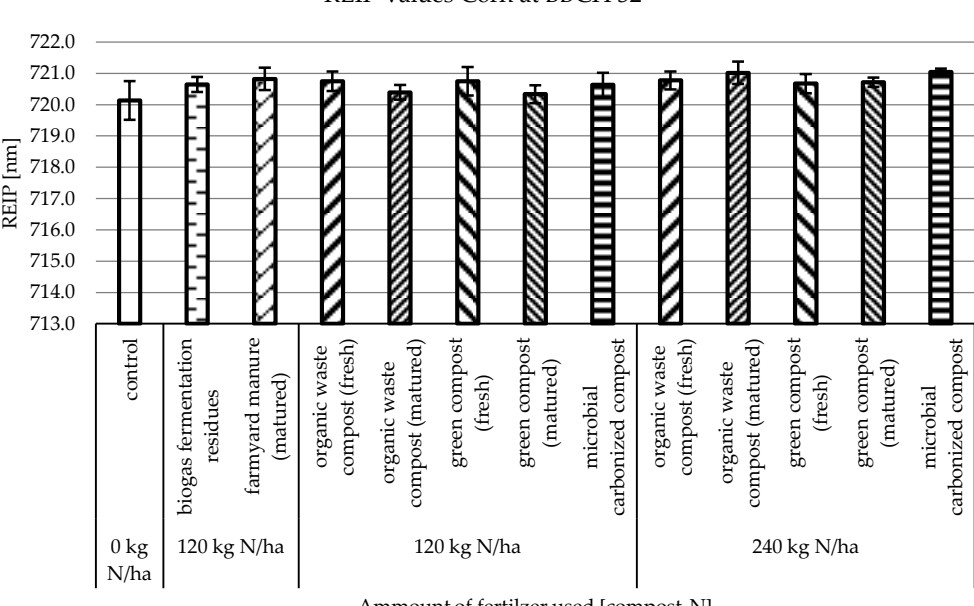

**Figure 5. Measured** REIP-values at BBCH 32.

The differences that were observed in growth stage BBCH 32 became clearer and statistically significant at growth stage BBCH 33 (Figure 6).

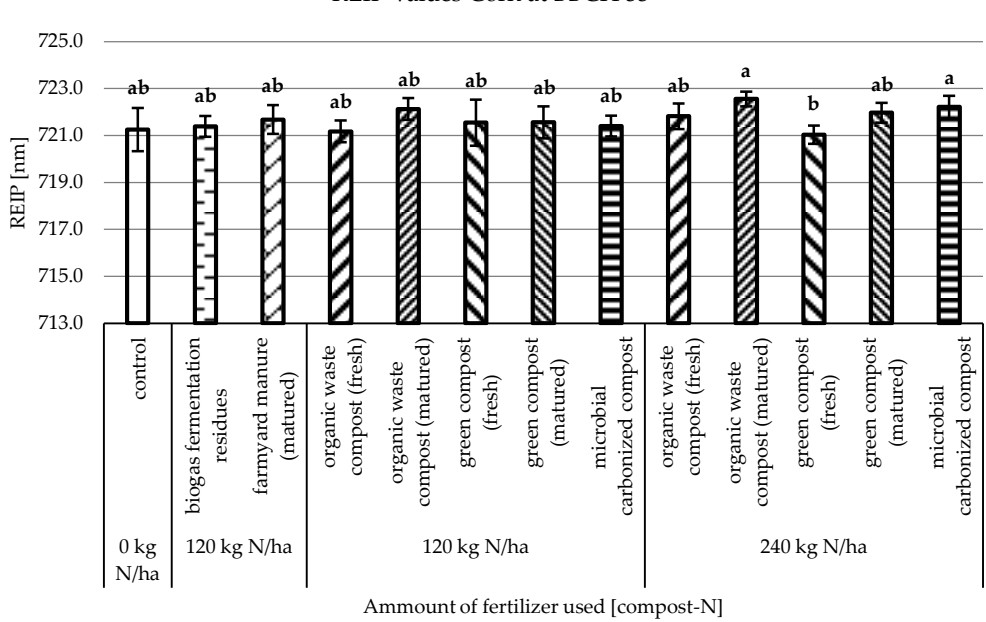

**Figure 6.** Measured REIP-values at BBCH 33 (variants with same letters do not differ significantly at 95% level, ANOVA with Tukey-b test).

The same development over time was also observed for the nitrate content in the soil (Figure 7).

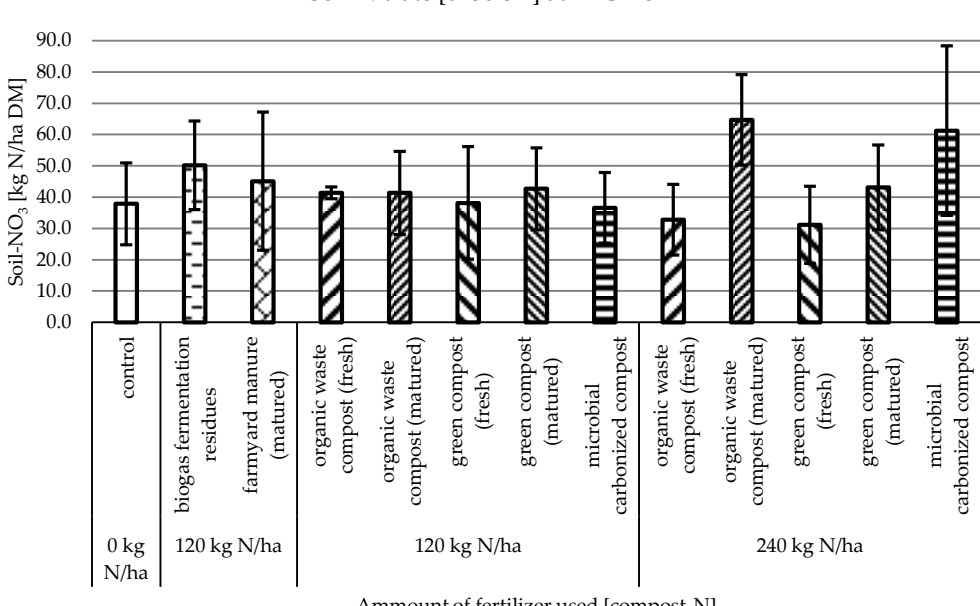

**Figure 7.** Measurement results of the soil-nitrate at growth stage BBCH 32.

## 4. Discussion

Measuring corn plants fertilized with organic fertilizers using field spectroscopy is much more difficult than other plants fertilized with mineral fertilizers due to the release of soil-borne N in combination with environmental circumstances such as the soil temperature and water supply through rainfall [52]. Nevertheless, it was possible to detect differences in the development of the plants. Similar to the findings of [53], in this experiment the sensor showed these differences in different values of the REIP vegetation index.

At the first measurement date at the early growth stage of BBCH 13 (Figure 3), the development of the above-ground biomass was very low. The visible soil had a slight effect on the measurement results taken at that time [54,55]. Similar to the findings of [56] for winter wheat, the results showed clear differences between the various organic fertilizer variants. It is known from previous studies that corn reacts to a differentiated supply of N at this vegetation stage [57–60]. This can be measured with multispectral sensors and the REIP vegetation index [61]. Moreover, at this early stage the fresh composts showed the highest REIP values of all fertilizer variants. Unlike matured compost, for fresh compost the composting time is much shorter. Fresh composts are generally not yet as stable as matured composts [62–64]. Both organic waste composts, as well as green composts, contain a larger proportion of easily decomposable organic matter that promotes soil life and mineralization within the soil [65]. This can explain the faster fertilization effect in corn and, therefore, the higher values of the REIP vegetation index. At growth stage BBCH 14 (Figure 4), these differences became significant. Except for the unfertilized control variant, the fertilization with biogas fermentation residues, as well as fresh green and fresh organic waste composts, at an application rate of 120 kg N/ha and the matured green compost at an application rate of 240 kg N/ha were statistically clearly distinguishable. On the other hand, the highest REIP values were found for the variants fertilized with biogas fermentation residues and fresh organic and fresh green waste composts at an application rate of 240 kg N/ha. Except for the control variant, these compost variants were not statistically clearly differentiated. At growth stage BBCH 32 (Figure 5), other values of the vegetation index REIP could be observed. The highest values were recorded in the variants fertilized with matured organic waste compost with an application rate of 240 kg N/ha and with microbial carbonized compost. Fresh composts and biogas fermentation residues had decreased. A possible explanation could be that between the very early growth stages and stem elongation at growth stage BBCH 32, more organically bound N was released

due to higher soil temperatures and water supply [52]. These differences became clearer at the next growth stage of BBCH 33 and were significant (Figure 6). The highest REIP values were observed for the variant fertilized with matured green compost at an application rate of 240 kg N/ha (Figure 6). The second highest values of the REIP vegetation index showed the variant fertilized with 240 kg N/ha of the microbial carbonized composts. The variants with the organic waste compost and the microbial carbonized compost could be statistically significantly differentiated from the variant fertilized with green compost. All three of them had an application rate of 240 kg N/ha (Figure 6). A first result was that the differences between variants fertilized with 120 kg N/ha and 240 kg N/ha were not very high at later growth stages. This result was supported by the values of the REIP vegetation index. The interaction of microbial activity within the soil, as well as rising temperatures in combination with soil water, caused a higher mineralization rate and, therefore, a better supply of nitrogen for the growth of the plants [66,67]. The soil samples, taken with the sensor measurement at the same time, showed that the values for nitrate within the soil, which are also relevant for nitrogen loss, i.e., leaching nitrogen into the groundwater, showed the same behavior as the values of the REIP vegetation index measured with the field spectrometer (Figure 7). These results showed that, like in conventional farming systems, in organic farming the danger of nitrogen loss is evident [68,69]. With this paper, we tried to apply sensor measurements to organic farming which are already in use in conventional agriculture. The results showed us that there is much unforeseen behavior in the interaction of fertilizer, soil, plants, and the weather. In common with conventional agriculture, we found that regardless of its form, the more nitrogen that is applied to a field, the higher the risk for an uncontrolled loss of nitrogen. However, as already described, this is location dependent. As there is very little research work available for sensor measurements in combination with vegetation indices in organic fertilizer use, and especially in organic farming, sensor measurements can, therefore, help any farmer to gather more information about the interactions of the environment and their location. The organic farmer, therefore, collects long-term information about their location and the applied organic fertilization in particular. However, it is not only the farmer who collects these data. With sensor measurements we have a tool to record and collect a lot of information on a very small scale. This information helps to develop a decision-support tool for the farmer in organic farming systems. Nevertheless, to develop such a system over a long-time period, a lot of sensor measurements have to be recorded, with a lot of soil and plant samples as well, which have to be gathered. Another source of information is the laboratory analyses of composts regarding their composition of nutrients, as shown in Table 1. With this information and the already existing knowledge of the optimal N supply for various plants from conventional farming fertilization trials, this goal can be reached. Although organic fertilizers, and in this case composts, are not uniform (neither a batch nor a subsample of them), an additional benefit is to have information about the average composition of their nutrients. All this information together enables us to develop a decision-support system for organic farmers to help them decide whether, when, which, and how much compost they should apply for their intended crop rotation to achieve the minimum environmental burden possible in combination with an optimal supply of nutrients for the plants.

## 5. Conclusions

A high application rate of nitrogen, regardless of its form, i.e., mineral or organic, always increases the risk of over-fertilization and, thus, a loss of nitrogen. The difference between a mineral and organic fertilization is the period of time at which nitrogen is available for the plants. In contrast to a conventional farmer, an organic farmer has much more difficulties in handling nitrogen. Field spectroscopy could help to learn more about the properties of their fields and to come to a clearer understanding of the interactions of the soil, fertilizer, climate, and plants. Using the REIP vegetation index enabled to identify small differences in the effect of the fertilized compost using the reflection of the sunlight

by the plants. This technique offers researchers and farmers new real-time large-scale potentials in organic fertilizing to gather more knowledge about the soil system, water, temperature, fertilizer, and specific crop. This could lead to a more efficient organic farming system, to achieve a higher yield and minimize the environmental burden at the same time.

**Author Contributions:** Writing—original draft, M.S.; Supervision, L.C. and K.-J.H. All authors have read and agreed to the published version of the manuscript.

**Funding:** The project ProBio (Optimum production and agronomic use of compost from organic and green waste in organic farming, Funding number [FKZ]: 2818OE009) is supported by funds of the Federal Ministry of Food and Agriculture (BMEL) based on a decision of the parliament of the Federal Republic of Germany via the Federal Office for Agriculture and food (BLE) under the Federal Program for Ecological Farming.

**Institutional Review Board Statement:** Not applicable.

**Informed Consent Statement:** Informed consent was obtained from all subjects involved in the study.

**Data Availability Statement:** The data in this study are available on request. They are not online available.

**Acknowledgments:** We would like to acknowledge all ProBio project partners and the producers of organic fertilizers in the experiment. We are grateful to the research station Viehhausen of the Technical University of Munich, especially Horst Laffert, Florian Schmid, and Stefan Kimmelmann, for their help with the management of the long-term experiment. Thanks to Johann Ludwig, Nicole Maier, and Isabella Hohenester for help with the field work.

**Conflicts of Interest:** The authors declare no conflict of interest.

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
