# Peer review of "Compost Fertilization in Organic Agriculture—A Comparison of the Impact on Corn Plants Using Field Spectroscopy"

_applsci, doi:10.3390/app13063676_

Round 1

Reviewer 1 Report

1. line 163 please check the word "re".

2. Figure 3、4、5、6、7, The Y-axis starting data starts at 0, and the differences between the treatments need to be marked. It is necessary to supplement what statistical methods were used to compare the differences between treatments. 

3. Merge the Table "Compost variants and significant ates at ***" with the corresponding Figure. The difference is more easily understood on the Figure.

4. Judging from the error lines between treatments, the differences between treatments seem insignificant.

Author Response

Reviewer 1:

  1. line 163 please check the word "re".

Checked and reformulated

  1. Figure 3、4、5、6、7, The Y-axis starting data starts at 0, and the differences between the treatments need to be marked. It is necessary to supplement what statistical methods were used to compare the differences between treatments.

Differences marked, statistical methods added

  1. Merge the Table "Compost variants and significant ates at ***" with the corresponding Figure. The difference is more easily understood on the Figure.

Tables merged with the corresponding figure

  1. Judging from the error lines between treatments, the differences between treatments seem insignificant.

Explanation of significance and insignificance added into the text

Reviewer 2 Report

The subject is interesting.

The result of the abstract is not impressive and should be modified.

lines 103-107: Mention the latitude and longitude of the studied area.

Line 131: This contrasts with the abstract section (line 13) you mentioned that “In a field trial established in 2017”

In the materials and methods section, mention what statistical method and statistical software you used for data analysis.

In the result section, for figures 3-7 for better understanding please Show the significant letters on the figures and remove tables 3 and 4.

The discussion is not well done. To the interpretation of those, you need to compare them with other studies. What is the strength or weakness of your results? What is your interpretation of them? Make your interpretation strong with other works. It’s an easy process. In discussion section, you just cited 1 reference!!

References are not up to date and should be updated. 

More specific comments are provided in the attached annotated pdf file.

Author Response

Reviewer 2:

  1. The subject is interesting.

Thank you very much!

  1. The result of the abstract is not impressive and should be modified.

Result of the abstract modified

  1. lines 103-107: Mention the latitude and longitude of the studied area.

Latitude and longitude added

  1. Line 131: This contrasts with the abstract section (line 13) you mentioned that “In a field trial established in 2017”

Details of the observation period added

  1. In the materials and methods section, mention what statistical method and statistical software you used for data analysis.

Statistical methods added and the statistical software mentioned

  1. In the result section, for figures 3-7 for better understanding please Show the significant letters on the figures and remove tables 3 and 4.

Figures and tables merged

  1. The discussion is not well done. To the interpretation of those, you need to compare them with other studies. What is the strength or weakness of your results? What is your interpretation of them? Make your interpretation strong with other works. It’s an easy process. In discussion section, you just cited 1 reference!!

Many new references added

  1. References are not up to date and should be updated.

References updated and added new ones

  1. More specific comments are provided in the attached annotated pdf file.

Thank you for the useful comments and hints! These were added into the text

Round 2

Reviewer 1 Report

Accept in present form.

Reviewer 2 Report

Dear authors,

 I think the present version is fine with me.

Hope for the best!!!

Reviewer